# Analysis of Chemical Composition and Biological Activities of Essential Oils from Different Parts of *Alpinia uraiensis* Hayata

**DOI:** 10.3390/molecules30071515

**Published:** 2025-03-28

**Authors:** Ying-Ju Chen, Fen-Hui Chen, Tse-Yen Liu, Yao-Moan Huang, Yi-Chiann Chen, Fu-Lan Hsu

**Affiliations:** 1Forest Products Utilization Division, Taiwan Forestry Research Institute, Ministry of Agriculture, Executive Yuan, Taipei 100051, Taiwan; yingju@tfri.gov.tw; 2Silviculture Division, Taiwan Forestry Research Institute, Ministry of Agriculture, Executive Yuan, Taipei 100051, Taiwan; fhchen@tfri.gov.tw (F.-H.C.); chiann@tfri.gov.tw (Y.-C.C.); 3Forest Protection Division, Taiwan Forestry Research Institute, Ministry of Agriculture, Executive Yuan, Taipei 100051, Taiwan; tyliu@tfri.gov.tw; 4Forest Ecology Division, Taiwan Forestry Research Institute, Ministry of Agriculture, Executive Yuan, Taipei 100051, Taiwan; huangym@tfri.gov.tw

**Keywords:** *Alpinia uraiensis*, Zingiberaceae, essential oil, hydrodistillation, leaf, flower, stem, rhizome, antioxidant activity, antifungal activity, *Phellinus noxius*

## Abstract

This study investigates the chemical composition and antioxidant and antifungal activities of essential oils (EOs) extracted from different parts of *Alpinia uraiensis* Hayata, an endemic species of Taiwan. EOs were obtained from flowers, leaves, stems, roots, and rhizomes using hydrodistillation, and their compositions were analyzed by gas chromatography–mass spectrometry (GC-MS). Monoterpene hydrocarbons and oxygenated monoterpenes were predominant, with flower and leaf EOs rich in γ-terpinene, terpinen-4-ol, and 1,8-cineole, while root EO was characterized by fenchyl acetate (72.2%). Antioxidant activity was evaluated using the DPPH assay, where the flower EO exhibited the highest radical scavenging activity (99.5% at 100 mg/mL), followed by rhizome and stem EOs, while root EO showed moderate activity. Antifungal activity against *Phellinus noxius*, a major wood-decaying pathogen, was assessed using the agar dilution method. Root EO exhibited the strongest antifungal effect, achieving complete inhibition at 400 and 800 μg/mL, while other EOs showed weaker activity. These findings suggest that *A. uraiensis* EOs possess potential antioxidant and antifungal properties, particularly the root EO, which may serve as a natural antifungal agent. Further studies are needed to elucidate the key bioactive compounds and their mechanisms for potential pharmaceutical and industrial applications.

## 1. Introduction

Zingiberaceae plants are key resources in tropical and subtropical regions; they are known for their remarkable diversity and extensive applications. *Alpinia* is a widely distributed, species-rich genus within the Zingiberaceae family; it comprises approximately 230 species that are distributed worldwide [1,2]. Taiwan has 18 species of *Alpinia*, including 12 endemic species [3], which highlights its rich biodiversity and regional characteristics. As a traditional ethnobotanical plant, *Alpinia* plays a crucial role in Taiwan’s culture and daily life. Various parts of the plant, including its leaves, flowers, fruit, stems, and rhizomes, have aromatic properties and are widely used in food packaging, food flavoring, and traditional medicine [4,5,6].

Over the past two decades, *Alpinia* species have attracted considerable attention in academic research because of their diverse chemical compositions and potential medicinal value [7]. Many studies have indicated that EOs extracted from *Alpinia* have potent antimicrobial [8,9,10], antioxidant [11,12], and anticancer [4,5,13,14] effects, indicating that they hold potential for drug development. Notably, the composition and yield of EOs extracted from different parts of *Alpinia* plants vary significantly. For instance, oils extracted from aerial parts, such as flowers, leaves, and fruits, are rich in camphor, α-terpineol, terpinen-4-ol, and 1,8-cineole, whereas oils extracted from underground parts, such as rhizomes, are rich in fenchyl acetate and 1,8-cineole [15,16,17,18]. Extraction of EOs from flowers has attracted particular attention because such extraction has a relatively high yield [19]. In addition, extraction of EOs from leaves is economically advantageous because of the abundant biomass of leaves, and EOs extracted from rhizomes have potential medicinal applications.

*Alpinia uraiensis* Hayata is an endemic species in Taiwan that has large flowers and an intense fragrance. This species is primarily found in Wulai in northern Taiwan. However, there is a dearth of research on the essential oil composition of this species. Moreover, systematic studies on the chemical composition of EOs extracted from different parts of *Alpinia* species in Taiwan are relatively scarce, which has limited further exploration of their bioactivity and potential applications. Among the various bioactivities of essential oils, their antifungal properties have garnered increasing attention. *Phellinus noxius* (Corner) Cunningham is a highly destructive wood-decay fungus responsible for brown root rot disease, which poses a significant threat to forest ecosystems and horticultural plants, particularly in tropical and subtropical regions [20]. Numerous studies have demonstrated that essential oils extracted from various plant species exhibit broad-spectrum antimicrobial activity against multiple bacterial and fungal pathogens [21]. Specifically, plant-derived natural compounds, particularly those rich in terpenoids, have shown promising antifungal effects, effectively inhibiting fungal growth [20,21,22]. Given the chemically diverse composition of *Alpinia* essential oils, evaluating their antifungal potential against *P. noxius* could offer a natural and environmentally sustainable alternative for disease management. In addition to antifungal properties, essential oils have been widely recognized for their antioxidant potential, which is crucial for their applications in food preservation, cosmetics, and pharmaceuticals [21,23]. Natural antioxidants, particularly those derived from plant essential oils, have been extensively studied as safer alternatives to synthetic antioxidants. Many essential oils from *Alpinia* species have been reported to exhibit strong antioxidant activities, primarily attributed to their high content of oxygenated monoterpenes and phenolic compounds [24,25,26]. However, the antioxidant potential of *A. uraiensis* essential oils remains unexplored. Evaluating the free-radical scavenging activity of essential oils extracted from different parts of *A. uraiensis* can provide valuable insights into their potential as natural antioxidants and expand their practical applications.

Therefore, the current study comprehensively analyzed the essential oil composition of different parts of *A. uraiensis*. The goal of this study was to explore the chemical diversity and potential applications of *A. uraiensis* and to provide scientific evidence that can contribute to the conservation and development of *Alpinia* resources in Taiwan. Furthermore, we examined their antioxidant activity to determine their efficacy in free-radical scavenging and evaluated the antifungal activity of these essential oils against *P. noxius* to assess their potential as natural antifungal agents.

## 2. Results and Discussion

### 2.1. Yields of Essential Oils

Analysis of the composition of EOs from different parts of *A. uraiensis* revealed a yield of 0.24% (*v*/*w*) from leaves, 0.62% (*v*/*w*) from flowers, 0.05% (*v*/*w*) from roots, 0.16% (*v*/*w*) from stems, and 0.37% (*v*/*w*) from rhizomes, with all calculated on a dry weight basis. All EOs were yellow. Compared with the essential oil yields of *Alpinia* species from other regions, the yields obtained from the flowers (0.62%) and rhizomes (0.37%) of *A. uraiensis* were considerably higher than those reported in previous studies (0.06–0.3% from flowers and 0.05–0.21% from rhizomes; Appendix A) [9,15,16,19,27,28,29,30]. By contrast, the yield obtained from roots (0.05%) was significantly lower than those reported in previous studies (0.15–0.25%) [19,28,31,32,33]. The yields obtained from leaves (0.24%) and stems (0.16%) were similar to those reported in other studies (0.03–0.41% from leaves and 0.01–0.15% from stems) [9,15,19,27,28,29,30,31,32,33,34,35].

### 2.2. Essential Oil Composition of Different Plant Parts

Table 1 presents the chemical compositions of the EOs extracted from different parts of *A. uraiensis*. The leaf EOs were primarily composed of monoterpene hydrocarbons (53.3% ± 2.3%) and oxygenated monoterpenes (45.9% ± 2.3%), with a small fraction of sesquiterpene hydrocarbons (0.7% ± 0.1%). Specifically, the major compounds in the leaf EOs were γ-terpinene (24.0% ± 3.6%), terpinen-4-ol (22.6% ± 0.9%), 1,8-cineole (16.8% ± 0.4%), and *p*-cymene (13.5% ± 0.3%). Additionally, the flower EOs were primarily composed of monoterpene hydrocarbons (52.7% ± 4.8%) and oxygenated monoterpenes (46.1% ± 5.0%), with a small fraction of sesquiterpene hydrocarbons (0.6% ± 0.4%). The major compounds in the flower EOs were terpinen-4-ol (28.9% ± 8.4%) and γ-terpinene (19.5% ± 0.6%), as well as α-terpinene (9.5% ± 0.2%), 1,8-cineole (8.8% ± 0.5%), and β-pinene (7.6% ± 2.2%). These results indicate that the composition of EOs from flowers was similar to that of those from leaves, with differences in the concentrations of terpinen-4-ol and γ-terpinene that may be attributable to the functional role of flowers (e.g., attraction of pollinators).

The essential oil composition from roots differed significantly, with predominant oxygenated monoterpenes (82.8% ± 3.3%) and minor monoterpene hydrocarbons (8.0% ± 1.5%). Sesquiterpene hydrocarbons (3.8% ± 0.9%) and oxygenated sesquiterpenes (0.7% ± 0.2%) were also detected. Fenchyl acetate, the main component of root EOs, accounted for a high proportion of 72.2 ± 3.5% and was not found in other parts, making it a characteristic component of the root. Similarly, fenchol (2.3% ± 0.0%), thymol methyl ether (2.2% ± 0.1%), and fenchone (1.2% ± 0.0%) were identified as characteristic components of root EOs.

The EOs extracted from the stems and rhizomes of *A. uraiensis* had similar chemical compositions. The oils from these two plant parts were predominantly composed of monoterpene hydrocarbons, with concentrations of 67.2% ± 0.4% in those from stems and 68.0% ± 2.1% in those from rhizomes. High amounts of oxygenated monoterpenes were also detected, accounting for 32.5% ± 0.4% of the composition of the oils from stems and 30.9% ± 1.9% of that of oils from rhizomes. By contrast, sesquiterpene hydrocarbons were present in limited amounts, accounting for 0.2% ± 0.1% of the composition of oils from stems and 0.9% ± 0.2% of that of oils from rhizomes. No oxygenated sesquiterpenes were detected. Overall, the major components in the stem and rhizome EOs were similar, with these components including γ-terpinene (22.8% ± 3.3% and 19.5% ± 3.6%, respectively), terpinen-4-ol (20.0% ± 3.0% and 16.9% ± 1.3%, respectively), β-pinene (15.0% ± 3.6% and 17.9% ± 1.3%, respectively), and α-terpinene (10.1% ± 2.9% and 8.4% ± 1.7%, respectively), all of which are monoterpene compounds. These results suggest that the stems and rhizomes of *A. uraiensis* share similar metabolic pathways for essential oil biosynthesis.

### 2.3. Comparative Analysis with Other Alpinia Species

Few studies have explored the compositions of EOs extracted from *Alpinia* species in Taiwan. The studies that have primarily included analyses of seed EOs from *A. zerumbet* (Pers.) B.L.Burtt & R.M.Sm. [36], seed and leaf EOs from *A. speciosa* [34], rhizome EOs from *A. officinarum* Hance [16], and leaf and rhizome EOs from *A. nantoensis* F.Y.Lu & Y.W.Kuo [15]. Although *A. uraiensis* is an endemic species of Taiwan, research into its chemical composition has been limited. Studies on this subject have specifically focused on analysis of ethanol extracts for their total phenolic content and antioxidant activity [37]. To the best of the authors’ knowledge, this is the first study to comprehensively examine the essential oil compositions of different parts of *A. uraiensis*.

#### 2.3.1. Leaf Essential Oil

Comparisons of *A. uraiensis* with other *Alpinia* species revealed major differences in essential oil compositions. For example, the major components of the EOs extracted from the leaves of *A. speciosa* have previously been identified as camphor (31.6%), sabinene (9.4%), γ-terpinene (8.0%), 1,8-cineole (5.6%), and terpinen-4-ol (5.3%) [34], while camphor (21.31%), camphene (11.41%), β-pinene (8.28%), *p*-cymene (5.25%), α-pinene (4.49%), and D-limonene (2.5%) were identified as the main EOs in the leaves of *A. nantoensis* (Appendix A) [15]. In the current study, the EOs extracted from the leaves of *A. uraiensis* were found to be rich in γ-terpinene (24.0%), terpinen-4-ol (22.6%), 1,8-cineole (16.8%), and *p*-cymene (13.5%), but contained only a small amount of camphor (2.2%).

Although camphor has been reported to hold potential in anti-inflammatory, antimicrobial, and insect-repellent applications [38], concerns regarding its toxicity and safety have limited its use in cosmetics, health products, and pharmaceuticals. By contrast, γ-terpinene [39,40], terpinen-4-ol [41], 1,8-cineole [42], and *p*-cymene [42] have been reported to have excellent antimicrobial properties. Additionally, terpinen-4-ol has been reported to confer various benefits for patients with cardiovascular disease and hypertension [36,43], and 1,8-cineole was identified as an effective topical anti-inflammatory, expectorant, and potent free-radical scavenger [42]. These findings suggest that the EOs extracted from the leaves of *A. uraiensis* can provide benefits in various applications. In contrast to other *Alpinia* species, *A. uraiensis* contains abundant bioactive compounds and has high safety and versatility for use. However, further research is required to explore the potential application of *A. uraiensis* in skincare, health products, and pharmaceuticals and evaluate its potential market value.

#### 2.3.2. Rhizome Essential Oil

This study observed notable differences between the compositions of EOs extracted from the rhizomes of three Taiwanese *Alpinia* species. As presented in Table 1, the main components of *A. uraiensis* rhizome EOs were identified as γ-terpinene (20.0%), β-pinene (17.9%), terpinen-4-ol (16.9%), 1,8-cineole (8.7%), and α-terpinene (8.4%). Although these compounds are present in other *Alpinia* species, their compositional proportions differ. For example, *A. nantoensis* rhizome EOs primarily contain camphor (29.58%), camphene (10.45%), and β-pinene (9.13%) [15], whereas *A. officinarum* rhizome EOs primarily contain 1,8-cineole (20.91%), 4-allylphenol acetate (12.50%), and β-bisabolene (6.18%) [16]. These compounds have strong potential in antimicrobial and antioxidant applications [12,15,16].

*Alpina nantoensis* rhizome EOs are rich in camphor and camphene, whereas *A. officinarum* rhizome EOs are rich in 4-allylphenol acetate and β-bisabolene. In the current study, these compounds were identified in low concentrations or were even absent in *A. uraiensis*. This discrepancy underscores the diversity of essential oil composition among species, which may be attributable to the genetic background, growth environment, and collection season of different species.

Additional comparisons were conducted with *Alpinia* species from Vietnam, Egypt, India, and Brazil, with these analyses confirming the distinctiveness of *A. uraiensis*. For example, EOs extracted from the roots of *A. uraiensis* were discovered to contain high proportions of fenchyl acetate (72.2%) and fenchol (2.3%), which are relatively scarce in other *Alpinia* species [17,18,32,44,45,46,47]. Notably, *A. mutica* Roxb., *A. macroura* K. Schum., and *A. strobiliformis* T.L. Wu & S.J. Chen from Vietnam contain high amounts of β-pinene (12.5–23.2%), 1,8-cineole (8.7–18.6%), and α-pinene (6.3–12.8%) [28,30,32], and *A. galanga* (L.) Sw. and *A. officinarum* from India contain high amounts of 1,8-cineole (2.1–55.39%) and fenchyl acetate (10.2–54.3%) [17,44,45,48].

In summary, high chemical diversity was noted in the composition of EOs from different parts of *A. uraiensis*. These differences stem from variations in secondary metabolites within the plant, resulting in unique components in the EOs from different parts. Therefore, investigating the EOs of individual plant parts is crucial, as it enables the identification of bioactive compounds specific to each part and facilitates the assessment of their potential applications. For example, Chen et al. indicated that *Alpinia guilinensis* fruit EO is rich in 1,8-cineole, while the leaves and stems contain different monoterpenes and sesquiterpenes, such as (+)-4-carene, terpinolene, and δ-cadinene, which are absent in the fruit or leaf EOs [35]. Such detailed analyses aid in the development of EO-based products targeting specific biological functions.

However, in addition to studying essential oils from individual parts, the comprehensive application of whole-plant essential oils also has significant advantages. From a functional perspective, whole-plant EOs, by combining components from different parts, may possess a broader spectrum of antibacterial activity [8,27]. For instance, the EOs from leaves and flowers may primarily provide antibacterial and antioxidant effects, whereas those from roots and rhizomes may contain anti-inflammatory or analgesic compounds. Whole-plant extraction not only ensures the retention of the most complete chemical composition but may also enhance the medicinal potential and multifunctionality of the EOs. The study by Chen et al. demonstrated that EOs from different parts of *A. guilinensis* exhibited significant antibacterial effects against various foodborne pathogens, including *Staphylococcus aureus*, *Bacillus subtilis*, *Escherichia coli*, and *Pseudomonas aeruginosa* [35]. This suggests that compounds from different parts may act synergistically, making whole-plant EOs more potent and broad-spectrum than single-part EOs, thereby enhancing their application value in medicine and food preservation. Additionally, whole-plant extraction improves resource utilization efficiency, reduces biological material waste, and further promotes sustainability.

In conclusion, this study provides a comprehensive analysis of the EO composition of different parts of *A. uraiensis*, contributing to the understanding of its chemical diversity. The findings underscore the importance of analyzing individual plant parts to better characterize their distinct chemical profiles, which may offer insights into their potential bioactivities. Additionally, the application of whole-plant essential oils demonstrates potential advantages, as synergistic interactions among different components may enhance their efficacy in medicine, food safety, and healthcare. Future research should focus on optimizing extraction techniques and investigating synergistic effects to maximize the application potential of *A. uraiensis*, while also promoting its sustainable utilization and conservation as a native plant resource in Taiwan.

### 2.4. Antioxidant Activity of Essential Oils from Different Parts of A. uraiensis

The DPPH free-radical scavenging activity of *A. uraiensis* EOs extracted from different plant parts was evaluated at various concentrations (Table 2). Among the tested EOs, the flower EO exhibited the highest DPPH scavenging activity, reaching 99.5 ± 0.5% at a concentration of 100 mg/mL, while the rhizome and stem EOs achieved free-radical scavenging rates of 86.4 ± 0.1% and 75.0 ± 0.9%, respectively. Due to the limited availability of the root EO, testing was initiated at the highest available concentration of 10 mg/mL, at which it exhibited a free-radical scavenging rate of 44.3 ± 1.4%. However, when compared with the positive control, ascorbic acid, under identical conditions (5 mg/mL), the EOs showed considerably lower activity. Overall, the free-radical scavenging efficiency among the different plant parts followed the trend flower ≥ root > rhizome > leaf > stem. In addition, the half-maximal inhibitory concentration (IC_50_) values of the EOs were estimated based on the curve fitting of the experimental data. The trend of IC_50_ values was as follows: flower (8.76 ± 0.20 mg/mL) < root (14.43 ± 1.34 mg/mL) < rhizome (25.04 ± 0.21 mg/mL) < stem (52.49 ± 2.02 mg/mL) < leaf (126.97 ± 9.58 mg/mL) (Table 2). It is noteworthy that the leaf and root EOs did not reach 50% scavenging activity at their highest tested concentrations. Therefore, their IC_50_ values were extrapolated from the fitted curves. Although extrapolated values may carry a higher degree of uncertainty, they still serve as useful references for comparing the antioxidant potentials of different plant parts. Overall, these findings are consistent with previous studies on essential oils from the *Alpinia* genus. For instance, Ghosh et al. reported that the rhizome EO of *Alpinia nigra* exhibited slightly superior antioxidant activity compared to that of other plant parts [27]. Furthermore, previous research on EO composition and antioxidant activity has demonstrated that EOs rich in oxygenated monoterpenes and phenolic compounds exhibit stronger antioxidant activity [49,50,51,52]. In the present study, the high radical scavenging ability of the flower EO may be attributed to its abundance of oxygenated monoterpenes, including terpinen-4-ol, β-pinene, α-terpinene, γ-terpinene, and 1,8-cineole. Among these components, γ-terpinene may play a particularly interesting role in antioxidant activity. Unlike typical radical scavengers such as phenols, γ-terpinene forms an unstable peroxyl radical upon hydrogen abstraction, which decomposes into *p*-cymene and hydroperoxyl radicals (HOO^•^). The latter can donate hydrogen to peroxyl radicals (ROO^•^), thereby terminating oxidative chain reactions and enhancing the EO’s antioxidant capacity [49]. The presence of terpinen-4-ol, 1,8-cineole, and β-pinene in the flower EO likely contributes to its radical-trapping capability. Terpinen-4-ol, for example, contains a hydroxyl (-OH) group, which can directly donate a hydrogen atom to neutralize peroxyl radicals [49]. Moreover, 1,8-cineole, despite having a lower antioxidant activity compared to phenolic compounds, may still contribute by stabilizing radical intermediates and modulating oxidative pathways. However, the leaf EO, which shares a similar composition with the flower EO, did not exhibit comparable antioxidant activity. This discrepancy may be due to the influence of other chemical constituents, as even minor components in EOs can contribute to antioxidant activity either directly or through synergistic interactions with other bioactive compounds [51]. The high antioxidant activity of the root EO is likely due to the presence of the phenolic compound thymol methyl ether. Phenolic compounds possess redox properties and function as reducing agents, hydrogen donors, and singlet oxygen quenchers [53]. Research has shown a strong correlation between the antioxidant activity of essential oils and their phenolic contents, with the cumulative contribution of individual phenolic compounds determining their free-radical-scavenging efficiency [24,51]. Accordingly, this study’s findings suggest that the root EO’s antioxidant potential is primarily driven by its rich phenolic composition and capacity to effectively scavenge ROO^•^ radicals, thereby mitigating oxidative damage.

Despite the moderate DPPH free-radical scavenging activity observed for *A. uraiensis* EOs, their antioxidant potential was lower than that of ascorbic acid, the positive control. This is in accordance with previous reports indicating that plant EOs generally exhibit lower radical scavenging activity compared to synthetic antioxidants or polyphenol-rich extracts [53]. Nevertheless, *A. uraiensis* EOs, particularly those from the leaves, hold potential as natural antioxidants for applications in food preservation, pharmaceuticals, and cosmetics. Further studies should investigate the synergistic effects of individual components and validate their bioactivity through additional in vitro and in vivo experiments to ascertain their application potential.

### 2.5. Antifungal Activity of Essential Oils from Different Parts of A. uraiensis

The antifungal effects of *A. uraiensis* EOs at different concentrations against *P. noxius*, along with their minimum inhibitory concentrations (MICs), are summarized in Table 3. The root EO exhibited the highest antifungal activity, achieving a 100% inhibition rate at both 400 and 800 μg/mL. Even at lower concentrations (200 and 100 μg/mL), the root EO maintained a strong inhibitory effect, with antifungal indices of 96.9% and 71.8%, respectively. Among the tested samples, only the root EO exhibited a clear MIC value of 200 μg/mL, while the other EOs showed no complete inhibition at 800 μg/mL and were recorded as > 800 μg/mL. These results indicate that while the root EO possesses promising antifungal potential, the overall inhibitory activity of *A. uraiensis* EOs is weaker compared to other essential oils or plant extracts previously reported for *P. noxius* control.

There is limited research on the antifungal effects of essential oils against *P. noxius*, as most studies have focused on plant extracts or isolated compounds rather than whole essential oils. For instance, Chen et al. isolated ferruginol, T-cadinol, α-cadinol, and T-muurolol from the heartwood extract of *Taiwania cryptomerioides*, which exhibited potent antifungal activity against *P. noxius*, with IC_50_ values of 16.9, 25.8, 33.8, and 50.6 μg/mL, respectively [54]. Similarly, Cheng et al. reported that essential oils from *Cinnamomum osmophloeum* leaves showed inhibitory effects against *P. noxius*, particularly those rich in cinnamaldehyde or a mixture of cinnamaldehyde and cinnamyl acetate, with IC_50_ values of 119.5 and 154.1 μg/mL, respectively [20]. Additionally, Hsiao et al. found that essential oil from *Cunninghamia lanceolata var. konishii*, particularly its major component cedrol, effectively inhibited *P. noxius*, with an IC_50_ value of 15.7 μg/mL [55]. Further mechanistic studies suggested that secondary metabolites in these oils may suppress fungal growth through multiple pathways, including disruption of fungal cell membranes, inhibition of cell wall synthesis, suppression of cell division, interference with protein synthesis, and mitochondrial dysfunction [55].

Compared to these previous reports, the antifungal activity of *A. uraiensis* EOs—aside from the root EO—is relatively weak. The lower efficacy may be attributed to differences in chemical composition, particularly the lower abundance of highly active phenolic or oxygenated sesquiterpenoid compounds in most parts of the plant [22,56]. Further analysis of the root EO’s composition may help identify the key antifungal compounds responsible for its superior activity. While the antifungal activity of *A. uraiensis* EOs is not as potent as certain previously studied plant extracts, their potential use as a natural antifungal agent still warrants further investigation.

## 3. Materials and Methods

### 3.1. Collection of Plant Material

Specimens of *A. uraiensis* were collected from Xinxian Nursery in northern Taiwan (24°50′27.2″ N, 121°32′1.5″ E) at an elevation of 341 m above mean sea level. Field collection was conducted during the flowering period, between March and April 2023. Healthy, well-grown specimens were harvested, with different plant parts, including flowers, leaf sheaths, stems, and roots, separately collected. Taxonomic identification was confirmed by Dr. Yen-Hsueh Tseng from the Taiwan Forestry Research Institute.

Two clumps of *A. uraiensis* were harvested as whole plants. Each clump yielded approximately 1–2 kg of flowers, 5–6 kg of leaves, 11–13 kg of stems, and 2–15 kg of rhizomes with fibrous roots. Of these plant parts, stems had the highest yield, followed by rhizomes, leaves, and flowers (Figure 1).

### 3.2. Isolation of Essential Oils

Freshly harvested samples (1 kg each) were subjected to hydrodistillation three times on a Clevenger apparatus for 4 h to isolate EOs from different plant parts. The resulting oils were dried by adding anhydrous sodium sulfate (Na₂SO₄) to remove any remaining moisture. Subsequently, the dried oils were transferred to airtight containers and stored at 4 °C to ensure their stability and preservation. Essential oil yields and experimental data were calculated from triplicate analyses, and the results are presented as means ± standard deviations.

### 3.3. Essential Oil Analysis

EOs were analyzed on a Clarus 600 gas chromatograph mass spectrometer (PerkinElmer, Waltham, MA, USA) equipped with a DB-5ms column (J&W Scientific, Folsom, CA, USA). A chromatographic temperature program was run as follows: from 50 °C to 120 °C at 4 °C/min, followed by heating at a rate of 3 °C/min to 180 °C and then at a rate of 35 °C/min to 250 °C, with a final holding step for 5 min. Helium was used at a flow rate of 1 mL/min as a carrier gas. The temperatures of the injection port and transfer line were both set to 250 °C. The mass spectrometer was operated at an ionization voltage of 70 eV and an ion source temperature of 230 °C. Data acquisition was performed in full-scan mode (30–400 *m*/*z*). AIs were calculated for all compounds with a homologous series of *n*-alkanes (C_8_−C_25_) on the DB-5ms column [57]. To confirm peak identities, mass spectra were compared with those in the NIST/Wiley and Adams mass spectral libraries, literature reports, and with those obtained from authentic compounds (Sigma, St. Louis, MO, USA).

### 3.4. DPPH Free-Radical Scavenging Assay

The DPPH free-radical scavenging activity of the essential oil was determined following the modified methods of Polatoglu et al. [58]. and Lin et al. [53]. The assay evaluates the antioxidant potential of the oil by measuring the decolorization of a 0.1 mM ethanol solution of DPPH.

Different concentrations of the essential oil (100, 50, 10, 5, 2.5, and 1.25 mg/mL) were prepared in ethanol, while ascorbic acid was used as a positive control at 5, 2.5, and 1.25 mg/mL. Due to the limited yield of essential oil from the root part, the starting concentration for this sample was set at a maximum of 10 mg/mL. In a 96-well microplate, 50 μL of each sample solution was mixed with 50 μL of 0.1 mM DPPH solution, along with positive and blank controls. The mixtures were shaken and incubated in the dark at room temperature for 60 min, and their absorbance was measured at 517 nm. Ethanol served as the negative control, and ascorbic acid was used as the standard antioxidant reference.

DPPH radical scavenging activity was calculated by using the following equation: Scavenging activity (%) = (1 − absorbance of essential oil/absorbance of control) × 100%. The results were given as the mean of three parallel experiments with standard deviation in Table 2. The half-maximal inhibitory concentration (IC_50_) was estimated by applying nonlinear curve fitting to the scavenging activity data across different concentrations of essential oils. The resulting IC_50_ values are presented in Table 2. For samples that did not reach 50% inhibition at the highest tested concentration, the values were extrapolated based on the fitted curves.

### 3.5. Fungal Strain

The brown root rot fungal strains *Phellinus noxius* (Corner) Cunningham Acpn003, Copn001, and Fbpn001 were isolated from *Araucaria cunninghamii*, *Cinnamomum osmophloeum*, and *Ficus benjamina*, respectively, by Dr. Tse-Yen Liu (Taiwan Forestry Research Institute). The Copn001 isolate was collected from a plantation in Nantou, Taiwan. The Acpn003 and Fbpn001 isolates were isolated from the diseased trees in Taipei, Taiwan. All the isolates were stored at the Taiwan Forestry Research Institute, Taipei, Taiwan.

### 3.6. Antifungal Assay

The antifungal activity of the EOs extracted from different parts of *A. uraiensis* was evaluated using the agar dilution method [59]. The EOs were dissolved in 150 μL of 99.5% ethanol and incorporated into 15 mL of sterilized potato dextrose agar (PDA) in 9-cm Petri dishes. Ethanol was used as a control. After the medium solidified, a 5-mm diameter mycelial plug of *P. noxius* was placed at the center of each dish and sealed with parafilm. The plates were incubated at 27 ± 2 °C and 70% relative humidity until the mycelia in the control group (without essential oils or constituents) reached the edge of the Petri dish. The antifungal index (AI, %) was calculated using the formulaAI (%) = (1 − *D*a/*D*b) × 100
where *D*a represents the diameter of the fungal growth zone in the experimental dish (cm) and *D*b represents the diameter of the fungal growth zone in the control dish (cm). The percentage inhibition of *P. noxius* growth at various concentrations and minimum inhibitory concentrations (MICs) value for each EO sample are presented in Table 3.

## 4. Conclusions

This study represents the first comprehensive analysis of EOs extracted from different parts of *A. uraiensis*, an endemic species of Taiwan. The results revealed that the yields and compositions of EOs significantly varied between different plant parts, with the oils having distinct chemical profiles. Specifically, the EOs extracted from the leaves and flowers were found to be rich in monoterpene hydrocarbons and oxygenated monoterpenes, such as γ-terpinene, terpinen-4-ol, and 1,8-cineole, whereas the EOs extracted from the roots were found to be rich in fenchyl acetate, a characteristic compound of this species. The EOs extracted from the stems and rhizomes were discovered to have similar chemical compositions, with the oils primarily composed of monoterpenes. Given the chemical diversity observed in different plant parts, utilizing the whole plant for EO extraction may provide a more comprehensive and functionally versatile composition, enhancing its potential applications in various industries.

Bioactivity assays demonstrated that *A. uraiensis* EOs possess moderate antioxidant activity, with flower EO exhibiting the highest DPPH free-radical scavenging capacity (99.5% at 100 mg/mL). Notably, the root EO exhibited strong antifungal activity against *P. noxius*, with a MIC value of 200 μg/mL. The potent antifungal activity of root EO suggests its potential as a natural antifungal agent, warranting further investigation into its bioactive constituents and mechanisms of action.

The distinct chemical and bioactive properties of *A. uraiensis* EOs highlight their potential applications in cosmetics and natural preservatives. Future studies should focus on identifying key active components, exploring synergistic effects, and evaluating their practical applications in sustainable plant-based antimicrobial and antioxidant solutions. These findings contribute to the conservation and industrial utilization of *A. uraiensis*, supporting the sustainable development of Taiwan’s endemic plant resources.

## Figures and Tables

**Figure 1 molecules-30-01515-f001:**
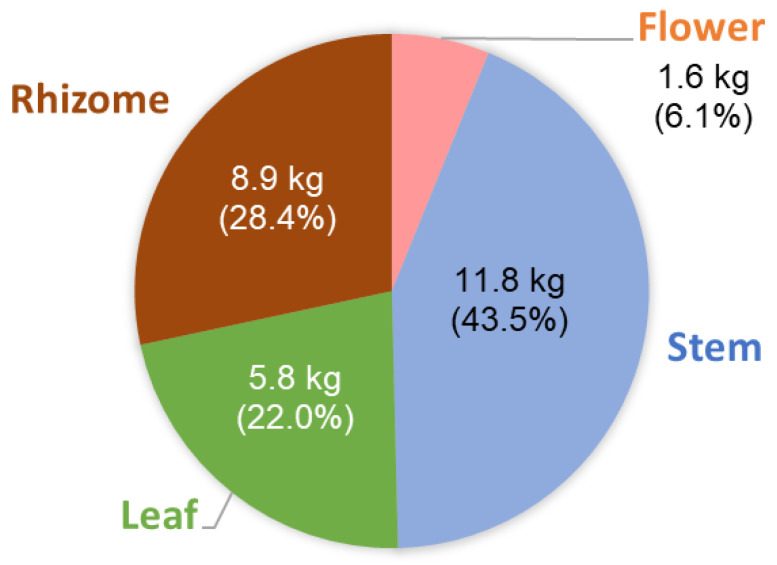
Biomass yield of different parts of *Alpinia uraiensis*.

**Table 1 molecules-30-01515-t001:** Chemical compositions (%) of essential oils extracted from *Alpinia uraiensis*.

Compounds	AI ^1^	rAI ^2^	Leaf	Flower	Root	Stem	Rhizome	Identification ^3^
2-Heptanol	902	894	2.3 ± 0.7 ^a^	0.5 ± 0.5 ^b^	n.d. ^4^	0.5 ± 0.5 ^b^	0.5 ± 0.2 ^b^	MS, AI
Tricyclene	924	921	0.0 ± 0.0 a	n.d.	n.d.	0.0 ± 0.0 ^a^	0.0 ± 0.0 ^a^	MS, AI
α-Thujene	927	924	0.3 ± 0.1 ^c^	0.9 ± 0.2 ^b^	n.d.	1.2 ± 0.1 ^ab^	1.4 ± 0.3 ^a^	MS, AI
α-Pinene	934	932	3.1 ± 0.8 ^bc^	1.9 ± 1.0 ^cd^	0.7 ± 0.5 ^d^	4.9 ± 1.1 ^ab^	5.7 ± 0.8 ^a^	MS, AI, ST
Camphene	951	946	1.0 ± 1.0 ^a^	0.9 ± 0.9 ^a^	1.2 ± 0.7 ^a^	1.0 ± 0.9 ^a^	1.1 ± 1.2 ^a^	MS, AI, ST
Sabinene	974	969	n.d.	3.4 ± 0.1 ^a^	n.d.	1.0 ± 0.2 ^b^	1.2 ± 0.2 ^b^	MS, AI, ST
β-Pinene	979	974	2.3 ± 0.9 ^c^	7.6 ± 2.2 ^b^	0.4 ± 0.2 ^c^	15.0 ± 3.6 ^a^	17.9 ± 1.3 ^a^	MS, AI, ST
Myrcene	990	988	0.8 ± 0.1 ^b^	1.3 ± 0.3 ^a^	0.4 ± 0.2 ^c^	1.2 ± 0.1 ^a^	1.2 ± 0.0 ^a^	MS, AI, ST
α-Phellandrene	1008	1002	0.3 ± 0.1 ^bc^	2.3 ± 1.0 ^a^	n.d.	0.9 ± 0.4 ^bc^	1.4 ± 0.6 ^ab^	MS, AI, ST
α-Terpinene	1018	1014	3.3 ± 0.9 ^b^	9.5 ± 0.2 ^a^	n.d.	10.1 ± 2.9 ^a^	8.4 ± 1.7 ^a^	MS, AI, ST
*p*-Cymene	1026	1020	13.5 ± 0.3 ^a^	0.8 ± 0.0 ^d^	1.0 ± 0.0 ^d^	3.8 ± 0.3 ^c^	5.4 ± 0.0 ^b^	MS, AI, ST
Limonene	1030	1024	2.8 ± 0.3 ^a^	1.1 ± 0.2 ^b^	2.6 ± 0.1 ^a^	1.6 ± 0.3 ^b^	1.7 ± 0.2 ^b^	MS, AI, ST
1,8-Cineole	1033	1026	16.8 ± 0.4 ^a^	8.8 ± 0.5 ^b^	2.9 ± 0.1 ^c^	8.5 ± 0.8 ^b^	8.7 ± 0.3 ^b^	MS, AI, ST
2-Heptyl acetate	1038	1038	n.d.	0.9 ± 0.1 ^a^	n.d.	n.d.	0.1 ± 0.2 ^b^	MS, AI
γ-Terpinene	1059	1054	24.0 ± 3.6 ^a^	19.5 ± 0.9 ^a^	1.1 ± 0.0 ^b^	22.8 ± 3.3 ^a^	19.5 ± 3.6 ^a^	MS, AI, ST
Terpinolene	1088	1086	1.8 ± 0.6 ^b^	3.3 ± 0.1 ^a^	0.5 ± 0.0 ^c^	3.7 ± 0.5 ^a^	3.0 ± 0.6 ^a^	MS, AI, ST
Fenchone	1090	1083	n.d.	n.d.	1.2 ± 0.0 ^a^	n.d.	n.d.	MS, AI
Linalool	1100	1095	0.9 ± 0.1 ^c^	2.0 ± 0.1 ^a^	n.d.	0.3 ± 0.1 ^d^	1.3 ± 0.1 ^b^	MS, AI, ST
Fenchol	1120	1118	n.d.	n.d.	2.3 ± 0.0 ^a^	n.d.	n.d.	MS, AI
Camphor	1149	1141	2.2 ± 2.3 ^a^	3.5 ± 3.7 ^a^	0.4 ± 0.1 ^a^	2.2 ± 2.3 ^a^	2.7 ± 3.0 ^a^	MS, AI, ST
Borneol	1173	1165	0.0 ± 0.0 ^b^	n.d.	0.8 ± 0.1 ^a^	0.0 ± 0.0 ^b^	0.0 ± 0.0 ^b^	MS, AI, ST
Terpinen-4-ol	1181	1174	22.6 ± 0.9 ^ab^	28.9 ± 8.4 ^a^	0.3 ± 0.1 ^c^	20.0 ± 3.0 ^ab^	16.9 ± 1.3 ^b^	MS, AI, ST
α-Terpineol	1196	1186	1.0 ± 0.1 ^bc^	1.5 ± 0.3 ^a^	0.2 ± 0.1 ^d^	1.1 ± 0.0 ^b^	0.7 ± 0.2 ^c^	MS, AI, ST
Fenchyl acetate	1219	1218	n.d.	n.d.	72.2 ± 3.5 ^a^	n.d.	n.d.	MS, AI, ST
Thymol methyl ether	1229	1232	n.d.	n.d.	2.2 ± 0.1 ^a^	n.d.	n.d.	MS, AI
Bornyl acetate	1284	1284	n.d.	n.d.	0.3 ± 0.1 ^a^	n.d.	n.d.	MS, AI, ST
α-Copaene	1375	1374	0.1 ± 0.1 ^a^	0.2 ± 0.2	n.d.	n.d.	0.2 ± 0.2 ^a^	MS, AI
β-Elemene	1389	1389	n.d.	n.d.	0.2 ± 0.0 ^a^	n.d.	n.d.	MS, AI, ST
(*E*)-Caryophyllene	1419	1417	0.1 ± 0.0 ^bc^	0.2 ± 0.1 ^b^	0.8 ± 0.1 ^a^	n.d.	0.0 ± 0.0 ^bc^	MS, AI, ST
α-Bergamotene	1432	1432	0.4 ± 0.1 ^ab^	0.2 ± 0.1 ^ab^	0.3 ± 0.1 ^ab^	0.2 ± 0.1 ^b^	0.4 ± 0.1 ^a^	MS, AI
α-Humulene	1454	1452	n.d.	n.d.	0.4 ± 0.1 ^a^	n.d.	n.d.	MS, AI, ST
4,5-di-*epi*-Aristolochene	1469	1471	n.d.	n.d.	0.6 ± 0.1 ^a^	n.d.	n.d.	-
Germacrene D	1488	1480	0.1 ± 0.1 ^a^	n.d.	0.2 ± 0.0 ^a^	n.d.	0.2 ± 0.0 ^a^	MS, AI
Aristolochene	1491	1487	n.d.	n.d.	0.2 ± 0.1 ^a^	n.d.	n.d.	MS, AI
β-Selinene	1495	1489	n.d.	n.d.	0.4 ± 0.1 ^a^	n.d.	n.d.	MS, AI
β-Dihydro agarofuran	1503	1503	n.d.	n.d.	0.2 ± 0.0 ^a^	n.d.	n.d.	MS, AI
β-Bisabolene	1508	1505	n.d.	n.d.	0.0 ± 0.0 ^a^	n.d.	0.0 ± 0.0 ^a^	MS, AI
γ-Cadinene	1511	1513	n.d.	n.d.	0.2 ± 0.1 ^a^	n.d.	n.d.	MS, AI
7-*epi*-α-Selinene	1517	1520	n.d.	n.d.	0.4 ± 0.1 ^a^	n.d.	n.d.	MS, AI
*trans*-Calamenene	1520	1521	n.d.	n.d.	0.3 ± 0.0 ^a^	n.d.	n.d.	MS, AI
Spathulenol	1579	1577	n.d.	n.d.	0.3 ± 0.1 ^a^	n.d.	n.d.	MS, AI
γ-Eudesmol	1620	1630	n.d.	n.d.	0.2 ± 0.1 ^a^	n.d.	n.d.	MS, AI
Monoterpene hydrocarbons			53.3 ± 2.3 ^b^	52.7 ± 4.8 ^b^	8.0 ± 1.5 ^c^	67.2 ± 0.4 ^a^	68.0 ± 2.1 ^a^	
Oxygenated monoterpenes			45.9 ± 2.3 ^b^	46.1 ± 5.0 ^b^	82.8 ± 3.3 ^a^	32.5 ± 0.4 ^c^	30.9 ± 1.9 ^c^	
Sesquiterpene hydrocarbons			0.7 ± 0.1 ^bc^	0.6 ± 0.4 ^bc^	3.8 ± 0.9 ^a^	0.2 ± 0.1 ^c^	0.9 ± 0.2 ^b^	
Oxygenated sesquiterpenes			n.d.	n.d.	0.7 ± 0.2 ^a^	n.d.	n.d.	

Only compounds with a relative abundance over than 1% in at least one column are listed. Different letters indicate significant differences between species (*p* < 0.05). Data are presented as mean ± SD (*n* = 3). ^1^ Arithmetic index (AI) relative to *n*-alkanes (C_8_–C_25_) on a DB-5ms column. ^2^ Reference AI. ^3^ Compound identified using AI, mass spectrum (MS), and standard compound (ST). ^4^ n.d.: not detected.

**Table 2 molecules-30-01515-t002:** DPPH scavenging activity (%) of essential oils from different parts of *A. uraiensis*.

Concentration (mg/mL)	Leaf	Flower	Root	Stem	Rhizome	Ascorbic Acid ^b^
100	48.6 ± 1.4 ^a^	99.5 ± 0.5	-	75.0 ± 0.9	86.4 ± 0.1	-
50	34.2 ± 0.7	97.5 ± 0.1	-	46.8 ± 1.0	68.3 ± 0.7	-
10	24.3 ± 1.3	53.5 ± 1.3	44.3 ± 1.4	20.4 ± 1.9	32.9 ± 0.5	-
5	17.5 ± 0.9	33.3 ± 1.2	30.8 ± 0.7	14.3 ± 0.6	20.9 ± 0.5	96.5 ± 0.1
2.5	11.3 ± 0.4	13.9 ± 0.2	23.1 ± 0.8	10.6 ± 0.3	12.6 ± 0.6	96.3 ± 0.2
1.25	6.1 ± 2.4	9.8 ± 0.7	16.2 ± 1.3	7.0 ± 0.4	7.2 ± 0.4	95.8 ± 0.3
IC_50_ (mg/mL)	126.97 ± 9.58	8.76 ± 0.20	14.43 ± 1.34	52.49 ± 2.02	25.04 ± 0.21	<1.25

^a^ Values are given as mean ± SD (*n* = 3). ^b^ Positive control.

**Table 3 molecules-30-01515-t003:** Antifungal activity of essential oils from different parts of *A. uraiensis* against *P. noxius*.

Concentration (μg/mL)	Antifungal Index (%)				
	Leaf	Flower	Root ^b^	Stem	Rhizome
800	33.3 ± 4.4 ^a^	56.0 ± 0.8	100.0 ± 0.0	28.6 ± 4.4	32.6 ± 4.6
400	26.4 ± 5.8	41.5 ± 3.0	100.0 ± 0.0	19.2 ± 6.8	23.9 ± 4.1
200	19.7 ± 5.2	29.6 ± 5.1	96.9 ± 5.4	17.2 ± 6.4	18.1 ± 1.7
100	16.1 ± 4.1	26.4 ± 3.6	71.8 ± 13.4	16.1 ± 4.7	16.1 ± 5.3
50	14.3 ± 5.0	16.5 ± 4.3	48.4 ± 15.5	11.4 ± 1.2	13.9 ± 1.6
MIC (μg/mL) ^c^	>800	>800	200	>800	>800

^a^ Values are given as mean ± SD (*n* = 3). ^b^ Values in this column were obtained using interpolation, with the original raw data presented in the Appendix A. ^c^ MIC was defined as the lowest concentration at which no visible fungal growth was observed.

## Data Availability

All data are presented within the manuscript.

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
