# Peer review of "Analysis of Chemical Composition and Biological Activities of Essential Oils from Different Parts of Alpinia uraiensis Hayata"

_molecules, 2025, doi:10.3390/molecules30071515_

Round 1
Reviewer 1 Report
Comments and Suggestions for Authors
The article entitled “Chemical Composition of Essential Oils Extracted from Different Parts of Alpinia uraiensis” presents a pleasant text, as well as its analysis of the data. It is somewhat unique, as it analyzes the constituents of essential oils extracted from different parts of the Alpinia uraiensis plant.
Most of the references used are current, and the quality of the tabulated data is good.
However, the article could be enriched by including experimental applications of the main constituents identified. In addition, a comparison between the different extracts from different parts of the plant could be made to evaluate these potential applications. These additional studies could increase the impact of the presented work.
Reviewer 2 Report
Comments and Suggestions for Authors
I read the manuscript 'Chemical Composition of Essential Oils Extracted from Different Parts of Alpinia uraiensis', analyzing essential oils from Alpinia uraiensis extracted from flowers, leaves, stems, roots, and rhizomes through hydrodistillation, and their chemical compositions identified using gas chromatography mass spectrometry. The introduction is quite short but sufficient, well presents the importance of essential oils from plants of the Alpinia genus and their bioactive properties.
I appreciate the incredibly detailed literature review of essnetial oils' compounds in other species of this genus (in the supplement).
I consider correct and reliable chromatographic analyses to be very valuable - with reference to the literature, use of numerous standards and comparison of calculated and literature retention indices.
I have some minor suggestions to improve your manuscript.
I suggest using some abbreviation for essential oils (eg. EO, EOs) as you used it near one hundred times and many times repeatedly in each subsequent sentence.
Table 1 - within the first 'n.d.' there is a '*' symbol but there's no explanation. Please add information if in table is given a standard deviation or standard error, etc. Also I don't understand why there are '± 0.0' after n.d. (if not detected it is impossible to count the value...). And one more thing: "Only components with a relative abundance greater than 1% are listed." but I see many compounds that are less abundant in each samples.
You can add to results, or to conclusions - maybe it is worth emphasizing the use of the whole plant for EO extraction in order to obtain the most complex composition for use in medicine?
I don't know if the article isn't a bit too short on analyses/results. It would be good to do some bioactivity, microbiology or at least antioxidant properties studies of individual essential oils to add a practical element, not just composition analyses, but I'll leave that to the Editorial Board to assess. From my point of view, the article is good, well written and contains no methodological errors, only minor issues.
